# Synthesis of Current Knowledge of the Morphology of the Larval Stages of Paederinae (Coleoptera; Staphylinidae), with a First Insight into the Mature Larva of *Pseudomedon* Mulsant & Rey, 1878, in the Light of a New Systematic Division

**DOI:** 10.3390/insects13110982

**Published:** 2022-10-26

**Authors:** Bernard Staniec, Ewa Pietrykowska-Tudruj, Grzegorz K. Wagner, Andrzej Mazur, Magdalena Kowalczyk

**Affiliations:** 1Department of Zoology and Nature Protection, Maria Curie-Skłodowska University, Akademicka 19, 20-033 Lublin, Poland; 2Department of Forest Entomology and Pathology, Poznan University of Life Sciences, 60-625 Poznań, Poland; 3Faculty of Biology and Biotechnology, Maria Curie-Sklodowska University, Akademicka 19, 20-033 Lublin, Poland

**Keywords:** staphylinids, immature stages, external structure, obsoletus, Medonina

## Abstract

**Simple Summary:**

The subfamily Paederinae is an example of a, species-wise, highly diverse group about whose developmental biology a great deal remains to be discovered, especially the external structures of their developmental stages. Here, we give the first morphological description of the mature larva of the genus *Pseudomedon* with detailed illustrations of its structural features. We have carried out a comparative analysis of these traits at different taxonomic levels on the basis of the new tribal system established for Paederinae and the current knowledge concerning the larval morphology of these staphylinids. We also provide an identification key for known Paederinae larvae and review in detail the relevant literature and the state of knowledge of their morphology.

**Abstract:**

This is the first morphological description of the mature larva (L2) of the genus *Pseudomedon*, belonging to the tribe Lathrobiini and subtribe Medonina. Detailed illustrations of its structural features are provided. Based on earlier published and new data, 10 and 18 diagnostic larval morphological characters for Paederinae and *Pseudomedon*, respectively, are proposed. In the light of the new tribal system established for the subfamily Paederinae and based on the current knowledge (including *Pseudomedon*) concerning the larval morphology of these staphylinids, a comparative analysis of the traits at different taxonomic levels was carried out: intertribal—between Lathrobiini and Paederini sensu nov., intersubtribal (Lathrobiini)—between Medonina and Lathrobiina, and intrasubtribal for Medonina. As a consequence, 12 intertribal, 2 intersubtribal and 3 intrasubtribal distinguishing features were selected. These features, appearing on the head, antennae and mouthparts of the larvae, confirm the validity of the recent proposals to alter the systematics of these staphylinids at higher taxonomic levels. Our proposed practical identification key to Paederinae larvae at the generic level is a synthesis of the current knowledge of Paederinae larvae, including new data. The work also gives a thorough review of the literature and the state of knowledge of the morphology of Paederinae larvae.

## 1. Introduction

Paederinae is one of the largest subfamilies of rove beetles (Staphylinidae and Coleoptera), containing around 7600 described extant species in 225 genera worldwide [1,2]. The majority of the studies relating to this group have mainly been taxonomic, in particular descriptions of new species or generic revisions, phylogenetic studies concentrating mainly on the morphological characters of adults, and to a lesser extent on larval stages, and very recently on both their morphological and molecular data, e.g., [1,2,3,4,5]. Unfortunately, the subfamily Paederinae remains an example of a, species-wise, highly diverse group about whose developmental biology a great deal remains to be discovered, especially the external structures of their developmental stages. 

### The State of Research on the Developmental Stages/Larval Stages of Paederinae

The larval morphology of the rove beetle subfamily Paederinae at various levels—species, genus or subtribe—despite a certain amount of the literature data concerning some 50 species (12 genera), has generally been poorly studied and remains an area that urgently needs examination (summary in Table 1). Just two recently published papers provide descriptive and graphical morphological data of the last-instar larvae that are sufficiently comprehensive and accurate for constructing a species identification key and for carrying out a phylogenetic analysis [6,7]. These papers relate to five species from four genera: *Lathrobium lineatocolle, Paederus littoralis, Paederidus rubrothoracicus carpathicola*, *P. ruficollis* and *Tetartopeus quadratus*, all from the European fauna.

Several papers, published mainly in the 1980s and 1990s by Ahmed [8], Watrous [9], Frania [3] and Smoleński [10], provide quite detailed morphological descriptions with illustrations that are generally reliable for diagnostics but not fully so for a phylogenetic analysis. The most comprehensive of these papers is the one by Frania [3], which focuses on the first-instar larvae of Stilicina (*Eustilicus* and *Rugilus*) and Medonina (*Deroderus*, *Stilocharis* and *Medon*), all except the first one (*E. fasta*) at the generic level. While this author highlights the considerable potential of first-instar larval characters for acquiring an understanding of the phylogenetic relationships in Paederinae, a substantial drawback of this work is the lack of summary tables for comparing the morphologies between genera or subtribes.

Some earlier morphological studies from the 1960s and 1970s, though graphically of poor quality, cover a wider range of species, each relating to over a dozen members of several different genera [11,12,13]. These fairly informative descriptions are quite useful for diagnostic purposes but have no value for phylogenetic analyses, as they focus only on the key diagnostic characters of species or genera while neglecting the remaining ones.

The most frequently described genus in the whole subfamily is *Paederus,* e.g., [7,14,15,16,17,18]. Data on the external larval structure of these brightly coloured, agile beetles are available for 14 species, most of them with a Palearctic or Holarctic distribution. However, such data are presented completely and illustrated satisfactorily for only two species—*P.*
*littoralis*, common throughout the Palearctic, and the Afrotropical *P. alfierii*; the morphological descriptions of the other species require a meticulous re-analysis.

Within the Medonina, a subtribe containing almost 50 genera, including *Pseudomedon* Mulsant & Rey, 1878, the larval morphological data are available for only 4 genera (*Deroderus, Lithocharis, Medon and Stilocharis*), usually represented by a single species [3,11,12,19]. Only those data describing the larvae of *M. johni* Blackwelder, 1943, are perhaps worthy of attention. This is a species with a globally very narrow distribution, being limited to Central America (Dominica, Saint John and St. Lucia) [20]. Whereas the description is fairly reliable, the graphics are unfortunately poor. All other data, even relating to such common and widespread species, such as the cosmopolitan *L. nigriceps*, extremely common throughout the Palaearctic, Oriental and Australasian regions, are solely of historical significance [11,19,21]. Accordingly, the highly detailed description given here of *Pseudomedon obsoletus* (Nordmann, 1837) larvae, enhanced by high-precision drawings, is the first such characterisation of morphological structures at the level of both the genus *Pseudomedon* and the subtribe Medonina. 

The genus *Pseudomedon* from the subtribe Medonina (Paederinae and Staphylinidae) currently contains 23 species. These are distributed mainly in the Palaearctic region, but there are also some in the Afrotropical and Australasian regions and a few in the Nearctic and Oriental regions. Most of the 13 species inhabiting the Palaearctic are known from the western parts of this region, including Central Asia; only two, recently described from India—*P. bengalensis* Assing, 2012—and from China—*P. schuelkei* Assing, 2011—have been reported from the Eastern Palaearctic. Among the Palearctic *Pseudomedon* fauna, there are certain proven generic affiliations, but the Afrotropical and Australian species are in a need of taxonomic revision. The sole Oriental species—*P. discolor* Assing, 2015—with an exceptional, conspicuous reddish-yellow body coloration, found in a primeval forest in northern Vietnam, is the most recently described member of the genus [21,22,23,24,25].

*Pseudomedon obsoletus* is widespread in the Western Palaearctic, from the Iberian Peninsula to Kazakhstan [23]. Like many species in the genus *Pseudomedon*, this species is very difficult to distinguish from congeneris solely on the basis of external characters, without an analysis of the male genitalia. Because up to the 1970s this species used to be identified only on the basis of its external appearance, which is very similar to that of the common *P.*
*obscurellus*, the two species were confounded. This confusing history of taxonomic interpretation within the genus *Pseudomedon* is described by Assing [23]. This small, dark rove beetle inhabits the wet margins of water bodies, moist meadows, rivers banks and lake shores, oxbows, meadows, peat bogs and forest margins and mixed deciduous forest with oak and floodplain forest. It lives under rotting plant remains, in flushes, among mosses and in the leaf litter, and during the winter, the beetle can be found by sieving reed litter and piles of straw. In Europe, this enigmatic species is rarely come across, usually caught singly, in greater numbers only at its wintering sites. In some parts of Europe, its population is regarded as distinctly declining [23,26].

## 2. Materials and Methods

### 2.1. Material Examined

Larval stages of *P. obsoletus* were obtained by rearing 4 adult specimens (Figure 1A–C). The beetles were collected by sifting wet leaf litter in the ecotone at the water’s edge of Lake Łukcze (Figure 1D) near the village of Rogóźno (51°23′29.32” N, 22°57′24.07” E; Łęczna-Włodawa Lake District, SE Poland) on 18 March 2019. Rearing took place from 20 March to 1 June 2019 at room temperature (20 ± 3 °C). Adults and larvae were kept separately in plastic containers (18 × 18 × 7 cm in height), filled with humid soil. Larvae and adults were fed with mites and Collembola of various species. The material for the morphological examination and measurements is listed in Table 2.

### 2.2. Study Techniques

Measurements (mm) of the first (L1) and second (L2—mature) larval instars were made with cellSens Dimension v1.9 software using an Olympus BX63 compound microscope. Photographs showing the overall habitus of the larvae (L2) and the male adult used for rearing were taken with an Olympus DP72 digital camera mounted on an Olympus SZX16 compound microscope.

To prepare the microscope slides for the morphological analyses, the preserved larvae were soaked in 10% KOH for several days, rinsed in distilled water, then immersed in lactic acid. Photographs showing the various details of the external structure of the larvae and male aedeagus were taken with an Olympus DP21 digital camera mounted on an Olympus BX63 compound microscope or with a VEGA3 TESCAN SEM and subsequently corrected using CorelDRAW Graphics Suite X6. The style and terminology of the morphological description are according to Staniec et al. [6,7].

The voucher specimens are deposited in the collection of the Department of Zoology and Nature Protection, Institute of Biological Sciences, Marie Curie-Sklodowska University, Lublin. The material used for the morphological examination and measurements (15 larvae—8 L1 and 7 L2) is listed in Table 2.

## 3. Results

### 3.1. Subfamily Diagnosis of the Mature Larvae (L2) of Paederinae

The combination of the characteristics that enable the mature larvae of Paederinae to be distinguished from the known larvae of other Staphylinidae are as follows (3, 6, 7, 10–13; 19, the present study): (1) body narrow, elongate, usually dorso-ventrally flattened, weakly sclerotized, small or medium-sized (usually 3–10 mm); (2) antennae four-articled, usually elongate and slim, articles III and IV clubbed, distinctly winding in the anterior portion; (3) antennal article III with three long setae and three solenidia—two long, one short; (4) each side of head with six oval or circular stemmata in clusters; exceptionally, head without stemmata; (5) stipes of maxilla, head and pronotum sides each with trichobothrium; (6) macro setae setose, micro setae (if they occur) match-shaped; (7) legs elongate and slim, sides of femur and tarsus almost parallel with long setae, setose; (8) article III of maxillary palp strongly tapering to top, distinctly longer than article II; (9) ligula elongate, tapering apically, pointed with tuft of microtrichia; and (10) urogomphi elongate, article I distinctly longer than segment X. Only 4 and 5 among the above characters are diagnostic solely of Paederinae; the others have been described in other, known staphylinid larvae. 

### 3.2. Generic Diagnosis of the Mature Larva (L2) of Pseudomedon

The diagnostic characters of the genus *Pseudomedon* are defined on the basis of the morphological data presented for the first time in this paper. The combination of the characters (indicated by arrows on the illustrations) that distinguish the mature larvae of *Pseudomedon* from the previously known larvae of the closely related Paederinae (6, 7, 11–13, 19, 30, 31) are the following: (1) head narrowest at around half-length; (2) head 2.5× wider than neck; (3) apotome reaching tentorial pits; (4) antennal article III longer than article II; (5) antennal article IV at most only slightly (1.2×) longer than sensory appendage of article III; (6) nasale without median tooth; (7) epipharynx with 22–25 triangular cuticular processes in anterior row; 8) mandible serrate with 15–17 teeth; (9) mala curved; (10) mala 3× as long as wide; (11) mala as long as article I of maxillary palp; (12) seta 2 of article II of maxillary palp tiny, located at about half-length of article; (13) microtrichia of hypopharynx form a vertical, median band; (14) article I of labial palp longer than ligula, at least twice as long as article II; (15) ligula at least 3× as long as wide at base; (16) urogomphi 4.3–4.4× longer (without apical seta) than pygopod; (17) article I of urogomphi about 2.5× as long as article II; and (18) article I of urogomphi with 1 club-shaped (coded: L3) seta.

### 3.3. Description

#### Morphological Description of Mature Larva (L2) of *Pseudomedon obsoletus*


Body moderately elongate and flattened; relatively well sclerotised; head almost as wide as pro- and mesonotum, slightly narrower than metanotum (Figure 2A,B). Coloration: head; antennae, maxillae, labium dark yellow and mandible red brown; pro-, meso- and metanotum brown with darker posterior and anterior edge, legs dark yellow; abdominal sclerites light brown, urogomphi and remaining parts of body almost colourless. Setae of different length—long pointed, short obtuse and light brown (Figure 2C,D1–D4).

Head (Figure 2E–G): rectangular in outline, sides rounded only in posterior part, widest at about ¼ length from base, narrowest at about half-length, then widening slightly to anterior margin; width ratio of neck and head in widest place = 1:2.5; moderately flattened dorso-ventrally, 1.3× as long as wide (in widest place); dorsal ecdysial lines (Es) bifurcate at about 0.4 length of head measured from anterior margin of nasale. Frontal part with five pairs of setae (coded: Fd1-2, Fl1-3); epicranial part (E) with 10 pairs of long setae (coded: Ed1-3, El1-3 and Em1-4), a pair of trichobothria (Trb); a pair of glands (Gl) and microstructure near Trb (Figure 2E,Ea). Chaetotaxy of ventral side with eight setae (2[V1-2, Vl1-2]); chaetotaxy of lateral sides with eight setae (L1-8) each (Figure 2F,G). Apotome (Ap) reaching tentorial pits (Tp) (Figure 2F,H). Posterior area (Pa) with gular region (Gr) as in Figure 2I,J. 

Functional positions of nasale (Na), antennae (At), mandibles (Md), maxillae (Mx), maxillary palps (Mp), labium (lb), labial palps (Lp) and ligula (Lg) as in Figure 3A,B. Antenna relatively short (Figure 3C,D): length ratios of articles I–IV = 1:2.7:4.1:2.4. Article I 1.7× as wide as long; article II almost 1.9× as long as wide, with two pores; article III 3× as long as wide, with three macro setae, one moderately elongate, finger-shaped and slightly curved sensory appendage (Sa), 4.6× as long as wide, three solenidia (IIIS1–3) of different lengths and two pores; article IV short and club-shaped, widening strongly to apex, 2.5× as long as wide, solenidia apically (IVS1–4); length ratio of article IV and Sa: 1.2:1. Stemmata present, ordered as shown in Figure 3E,F. Anterior margin of nasale with 8 obtuse teeth and 10 peg setae (Figure 3G–I)—four pairs of teeth (2[Pmt, Lt1-3]): a pair of wide paramedian teeth (Pmt), three pairs of narrow lateral ones (Lt1-3), Lt1 and Lt2 similar in size, Lt3 smaller than the others and median tooth absent; five pairs of peg setae (2[Ps1-5]): setae of Ps1-2 tiny and poorly visible, setae of Ps1 placed very close to each other, setae of Ps3-5 elongate and well visible. Epipharynx (Figure 3H) with two transverse rows of triangular cuticular processes; anterior (Acp) with 22–25 cuticular processes along anterior margin of buccal cavity (Bc); posterior (Pcp) with 12–13 cuticular processes along posterior margin of Bc incomplete medially and two pairs of lateral pores; pharynx (Ph) with a pair of sensilla (Sm). 

Mandible (Figure 4A,B) falciform, moderately elongate and narrow; serrate with 15–17 teeth; with two lateral setae (tiny [L1] and long [L2]), a large sensillum dorsally (Dsm) and two small sensilla near apex (Asm); seta L2 distinctly shorter than distance from seta to base of mandible. Maxilla (Figure 4C): length ratios of cardo (Cd) and stipes (Stp) = 1:1.1; cardo triangular in outline, 1.3× as long as wide with one seta dorsally; stipes rectangular in outline,1.9× as long as wide, with five setae, one ventrobasal trichobothrium (Trb) and two pores; mala (Ma) finger-shaped, relatively stocky, slightly curved, rounded apically, 3.1× as long as wide, with six setae different lengths (one basally, others near apex), one tiny sensory appendage apically and two pores basally (Figure 4C). Maxillary palp moderately elongate (Mp): length ratios of articles I–III = 1:1.1:2.4, respectively; article I and mala almost equal in length and width, 2.3× as long as wide with two pores apically; article II 2.6× as long as wide, slightly widening to apex, with two setae different lengths (coded: 1, 2), seta 1 long—located near apex, seta 2 short—located almost at half-length of article (Figure 4C); article III tapering to apex, 6.3× as long as wide with one digitiform sensory appendage basally, one seta close to apex and six elongate sensory appendages apically (Figure 4D). Labium (Figure 4E,Ea): ventral side—prementum (Pmnt) slightly extended anteriorly, rear portion strongly sclerotised; with four setae (two short, two long), two pores anterior and bunches of microtrichia anteriolaterally; dorsal side (hypopharynx)—membranous, with over a dozen microtrichia forming vertical, median band (Figure 4Ea). Labial palp (Lp) two-segmented (Figure 4E): length ratio of articles I and II = 2.1:1, article I slightly curved, 3.4× as long as wide with two pores near apex, article II 2.6× narrower than article I, 3.9× as long as wide, with bunch of cuticular processes (Cp) of different length apically (Figure 4F). Ligula (Lg) elongate and narrow, 3.1× as long as wide at the base, with numerous long microtrichia (Figure 4G), length ratio of ligula and segment I of Lp = 1:1.3; ligula and prementum completely separated by well-visible sclerotised strip (Sts).

Thorax. Fore leg (Figure 5A,Aa): coxa (Cx) moderately elongate) about 2.8× as long as wide, with 21 setae (Ad1–5, Al1–6, Av1–2, D1–3, Pl1, Pd1 and V1–3) and a few pores; trochanter (Tr) divided by row of a few sensilla (Sm), 3.3× as long as wide, with 6-8 setae and a few pores; femur (Fe) elongate, 5.5× as long as wide, with 36 setae (Ad1–10, Al1–4, Av1–6, D1–9, Pv1–3 and V1–4) and one pore; tibia (Tb) elongate and extremely slender, 10.4× as long as wide, with 22 setae (Al1–2, Av1–4, D1-3, Pd1–7, Pl1-2 and Pv1–4); tarsungulus (Ts) relatively short, slightly curved (Figure 5A,Aa), about 4.0× as long as basal width, with two spine-shaped setae (V1, Pl1), about 2.3× as long as seta V1. Length ratios of coxa, trochanter, femur, tibia and tarsungulus: 6.7:2.5:6.9:5.1:1, respectively. Thoracal tergites with clearly visible mid-longitudinal ecdysial line and transverse carina anterior and posterior, about equal in width (Figure 5B,C); pronotum only slightly shorter than meso- and metanotum combined, 1.2× as long as wide, moderately rounded on sides, widest at level of setae L6, with 15 pairs of setae (coded: A1–2, Da1–2, Db2, L1–6 and P1–4), a pair of trichobothria (Tb), a pair of tergal glands (Tg) and at least 11 pairs of pores (Ap1–3, Dp1-3 and Pp1-5). Mesonotum (Figure 5C): 1.9× wider than long, distinctly rounded on sides, widest at about half-length, with transverse carina anterior and 14 pairs of setae (coded: A1–5, Da2, Db2, L4–L6 and P1–4), a pair of tergal glands (Tg) and 5 pairs of pores (Ap1-2, Pp1-3); carina in the posterior part with transverse rows of long cuticular processes (Figure 5D,E). Chaetotaxy of metanotum similar to that of mesonotum. Prosternal region (Figure 5F,G): presternum (Pr) narrow, divided by medial longitudinal line, with four pairs of setae (one long) and one pair of pores; sternum (St) of prothorax transverse, divided into two sclerites, each wider than long, each with three short setae and a band of microtrichia on the outer margin (Figure 5F,H), sternites widely separated by membranous area (about 1/3 of sternite width). Ventral side of meso- and metathorax membranous, each with a pair of tiny setae (Figure 5G). Region between pro- and mesothorax with large, elliptical spiracles (Sp) and one dorsal seta near spiracle (Figure 5I,J).

Abdomen. Tergites and sternites of abdominal segments I-VIII divided by longitudinal membranes (Figure 6A–D). Segment I: tergite with transverse carina anterior and five pairs of setae (coded: A1, A3, L5—short, P2 and P4—long), a pair of tergal glands (Tg) and two pairs of pores (Pp1 and Pp2) (Figure 6A); paratergite (Pt) well developed on each side with two setae of different length, parasternites and sternite absent. Segments II-VIII (Figure 6A–E): tergite (Figure 6A,B) with 11–12 pairs of setae (coded: A1, Db2, L3-5, P1-2 and P4; uncoded: 3–4 club-shaped additional) and 1–2 pores (Pp1-2); sternites (St) well developed with 6—St II or 6-7 pairs of setae—St III-VIII (coded: A3, P1 and P4 (Figure 6C,D); uncoded 3-4 club-shaped additional); paratergites (Pt) and parasternites (Ps) elliptical and elongate on each side, with two and four setae (two long, well visible), respectively (Figure 6E). Abdominal segments I–VIII, each with a pair of conical spiracles (Sp) located between tergites and paratergites (Figure 6E,F). Segment IX (Figure 6B,D): without paratergite and parasternite, tergite with four pairs of setae and two pair of pores, sternite with five pairs of setae (two short). Urogomphus (Ug) two-articled: article I very elongate; tapering to apex, about 6.5× as long as wide at the base, with 12 setae—11 simple, pointed (coded: D1-D4, L1-2, L4, Lv1-2 and V2-3) and 1 club-shaped (coded: L3) (Figure 6G); article II very slender and short, about 8.3× as long as wide, about 3.4× narrower than segment I at the base, with one long apical seta and one short subapical seta (Figure 6G); length ratio of articles I, II and apical seta of Ug: 2.5:1:1.6, respectively; Ug 4.3–4.4 as long (without apical seta) as pygopod. Segment X (pygopod) tube-shaped: 1.7× as long as wide at base, with 11–12 pairs of setae (7 macro setae, well visible, 4–5 micro) and 2 pairs of pores (Figure 6D,Ga).

## 4. Discussion

### 4.1. Larval Characters at the Tribal Level in Paederini and Lathrobiini Sensu Novo 

The higher classification of the subfamily Paederinae has quite a long history, which stretches back to the first half of the 19th century. The changes in the classification systems of this group of staphylinids was described in detail by Frania [3], Schomann and Solodovnikov [2] and in abbreviated form by Żyła et al. [1]. The Paederinae taxonomy currently recognises four tribes—Pinophilini, Cylindroxystini, Paederini and Lathrobiini—instead of the two previously recognised ones—Pinophilini and Paederini. In this new tribal system, established on the basis of molecular data but supported by adult morphological data, the former tribe Paederini, apart from Cylindroxystini *stat. resurr*., was split into Paederini *sensu novo* and Lathrobiini *sensu novo* [2,34].

We assembled larval morphological data sampled from sufficiently detailed, current descriptions of species or genera of Paederinae and utilised them to assess their usefulness for confirming the validity of new tribal arrangements for two tribes, i.e., Lathrobiini and Paederini *sensu novo*. We did not analyse the other two tribes, i.e., the Neotropical Cylindroxystini and Pinophilini, as we are unfamiliar with their preimaginal stages. The characters common to Lathrobiini *sensu novo* presented in Table 3 are based mainly on those given by Staniec et al. [7] and the present study for the larvae of the subtribe Lathrobiina (*Lathrobium, Pseudomedon* and *Tetartopaeus*). In addition, certain data from Frania [3] have been taken into consideration for Stilicina (*Rugilus* and *Eustulicus*) and Medonina (*Medon*) and in the case of Paederini *sensu novo*, data given by Staniec et al. [6,7] and limited data from Kasule [12] and Pototskaya [11] for the subtribe Paederina (*Paederus* and *Paederidus*). This analysis has shown that the characters of Paederinae larvae perfectly match and wholly justify the new tribal arrangement of Lathrobiini and Paederini. Of course, it is well to bear in mind that the present state of knowledge of this group of rove beetles is still generally very unsatisfactory. Among the subtribes Paederina, Cryptobiina, Dolicaonina and Dicaxina in Paederini *sensu novo*, the larvae of only the first mentioned have been used for tribal diagnoses, the limited and confusing diagnoses of the second one given by Pototskaya [11] are poorly informative, while nothing is known about the other two. Likewise, within Lathrobiina, only the sets of characters established for Lathrobiina, Medonina and Stilicina could be extrapolated, because the other four—Scopaeina, Astenina, Stilicopsina and Echiasterina—are unknown.

Frania [3], in the study of the morphology of *Medon* in the subtribe Medonina and of some genera in other subtribes, i.e., Stilicina, Scopaeina, Stilicopsina and Astenina (all members of Lathrobiini *sensu novo*), notes that the larvae share a number of derived characters, which indicates that these subtribes are closely related. That author considers the following three characters to be the most important ones: (i) swollen, small setae (Fl2) on the nasale, (ii) branched microtrichia on the ligula and (iii) cuticular processes on the anterior margin of the buccal cavity arranged in rows. By contrast, Frania [3] considers that the larvae of the genera in the subtribes Paederina (now Paederini *sensu novo*) have the primitive condition for these three characters. However, the morphology of the larva of another member of the Medonina subtribe (*Pseudomedon*) analysed in the present paper does not uphold the hypothesis that the above-mentioned characters are common to this subtribe. Our observations show that the larvae of *Pseudomedon* do not possess a seta (F12) of a different structure on the nasale; it is of the simple type, and likewise, the microtrichia on the ligula also have a simple, not bifurcate, structure. Only the third of the three characters given by Frania [3] is valid for Medonina, but it is not, as assumed, in opposition to Paederini. However, it is difficult to give a firm, unequivocal response to Frania’s [3] conclusions, because the author did not identify the species of the analysed larvae, remaining on the generic level. It may be that characters (i) and (ii), analysed above, are intra- or intergenerically variable. Moreover, the specimens examined by Frania [3] were first-instar larvae: although these are similar to the second (final) instars, they are not identical to them. That is why transferring the morphological characters of larvae from the known genera of Paederinae worked out on the basis of model L2 to members of L1 from the Frania [3] paper is unreliable and may lead to erroneous inferences.

### 4.2. Comparison of the Larval Characters of Lathrobiini Sensu novo at the Inter- and Intrasubtribal Level

The detailed description of the morphological structures of the *Pseudomedon obsoletus* L2 larvae given in this paper is the first thorough elaboration of Medonina at both the generic and subtribal levels. Table 4 presents an inter- and intrasubtribal morphological comparison of the tribe Lathrobiini *sensu novo* based on our data and those gleaned from the literature relating to the four genera (3 to a limited extent, 7, 10]. The following inferences can be drawn from the data contained in this table. Among other known Lathrobiini *sensu novo,* the principal characters unique to the subtribe Medonina are: (i) the absence of micro setae, (ii) the small (the smallest in this tribe) number of macro setae on the surface of the epicranium and iii) the smooth surface of the epipharynx, with no microstructure. At the subtribal level, the labrum and maxilla are very useful structures, exhibiting intertribal similarities and differences. The distinct differences between Medonina-*M* (*Pseudomedon*) and Lathrobiina-*L* (*Lathrobium* plus *Tetartopaeus*) are: (i) the median tooth on the labrum—absent in *M* but present in *L*—and (ii) the mala relative to article I of the maxillary palp—no shorter in *M* and shorter in *L*. These characters link Medonina (*Pseudomedon*) with Stilicina (*Rugilus*), but these subtribes are distinguished by the apotome, which is a distinctly separate structure in the former but absent in the latter.

### 4.3. Key to Genera of the Known Larvae of Paederinae 

The identification key for the Paederinae larvae given below, encompassing eight genera, was constructed on the basis of the species inhabiting the Palearctic region. The key includes the previously undescribed larvae of *Pseudomedon*, the earlier well-known *Paedridus*, *Paederus*, *Lathrobium* and *Tetartopeus* [6,7], and *Medon*, *Ochtephilum* and *Rugilus*, known from incomplete descriptions [3,10,12]. Unfortunately, the morphologies of a large number of paederine larvae are still too poorly recognised to include them in this key (see Table 1). Here, we attempt to differentiate the taxa using mostly simple and user-friendly characters, the analysis of which does not require any prior preparatory effort.

1Sensory appendage of the antenna on the outer side of segment III …………… ..……………………………………………………….………….……….……***Ochtephilum***
-Sensory appendage of the antenna on the inner side of segment III …….….………….22Head rounded, buccal cavity wide, ligula conical …….………………….………………3…………………………………………….………………………….***Paederus***, ***Paederidus***
-Head of a different shape, buccal cavity narrow, ligula bulbiform or pear-shaped …………………………………………………………………..………….……….……..4…………………………….…***Lathrobium, Tetartopeus***, ***Medon***, ***Pseudomedon***, ***Rugilus***3Prosternum, tergite and sternite of abdominal segment VIII uniform, tuft of cuticular processes on epipharynx short, 2/3 of outer margin of mandible toothed ………………………..……………………………………………………………………….……….….. ***Paederidus***
-Prosternum, tergite and sternite of abdominal segment VIII divided, tuft of cuticular processes on epipharynx long, 1/3 of inner margin of mandible toothed .……………….….………………………………………………………………………………………..***Paederus***4Central part of anterior margin of nasale concave, mala shorter than article I of maxillary palp ……………………..……………………………………………………….…………..5……………………………………………….……………..***Pseudomedon***, ***Medon***, ***Rugilus***
-Central part of anterior margin of nasale distinctly conical, mala not shorter than article I of maxillary palp ……………………………………………………………….……….6……………………………………………………………..……..***Lathrobium***, ***Tetartopeus***5Apotome present, ligula wide at base (wider than base of article I of labial palp), no channel on epipharynx …..……..…………………………………………………………..…….7…………………………………………………………………………***Pseudomedon***, ***Medon***
-Apotome absent, ligula narrow at base (narrower than base of article I of labial palp), many channels on epiphaynx …………….…………..………..…….………………….***Rugilus***6Neck wide, median tooth of nasale narrow, mandible with inner margin serrate ……………………………………………………..……………………….…….***Tetartopaeus***
-Neck narrow, median tooth of nasale wide, mandible with inner margin smooth …………………………………………………………………………………***Lathrobium***7Single row of cuticular processes along anterior margin of buccal cavity in central part…………..……………………………………..…………………..……..…..…***Pseudomedon***
-A few rows of cuticular processes along anterior margin of buccal cavity in central part………………………………………………………………………..….…………..…***Medon***


## 5. Conclusions

The diagnostic characters of the Paederinae larvae at different taxonomic levels, given in this paper, are in accordance with the current state of knowledge and based on the limited larval material of the European fauna. Our intention is to expand these studies of the developmental stages of this rove beetle subfamily. Establishing new and certain (i.e., constant, valid) morphological data of these larvae at the species level will not only enable or facilitate them to be distinguished but also allow them to be included in phylogenetic studies, which to date have been conducted solely on the basis of imaginal and molecular data.

## Figures and Tables

**Figure 1 insects-13-00982-f001:**
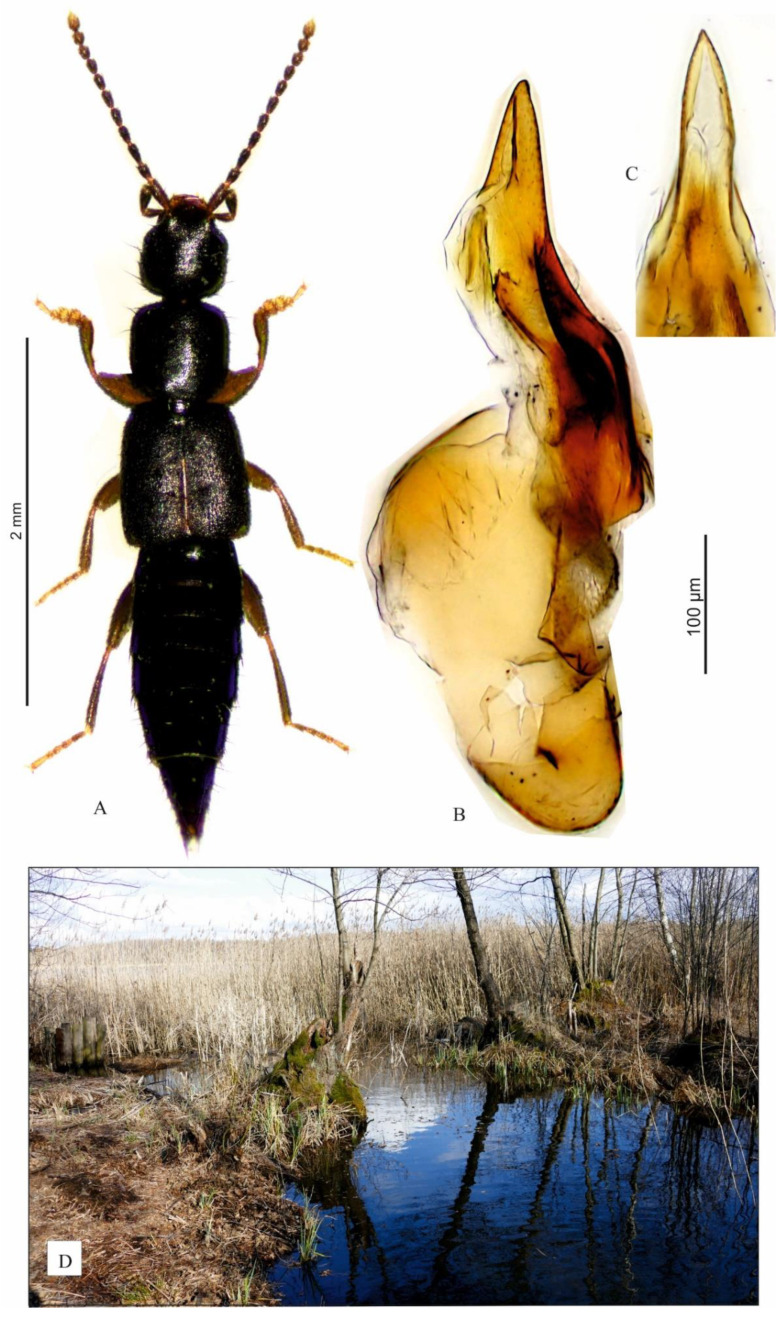
*Pseudomedon obsoletus*. (**A**), habitus of imago; (**B**,**C**), aedeagus in lateral aspect (**B**) and its apex (**C**) in ventral aspect; (**D**), biotope.

**Figure 2 insects-13-00982-f002:**
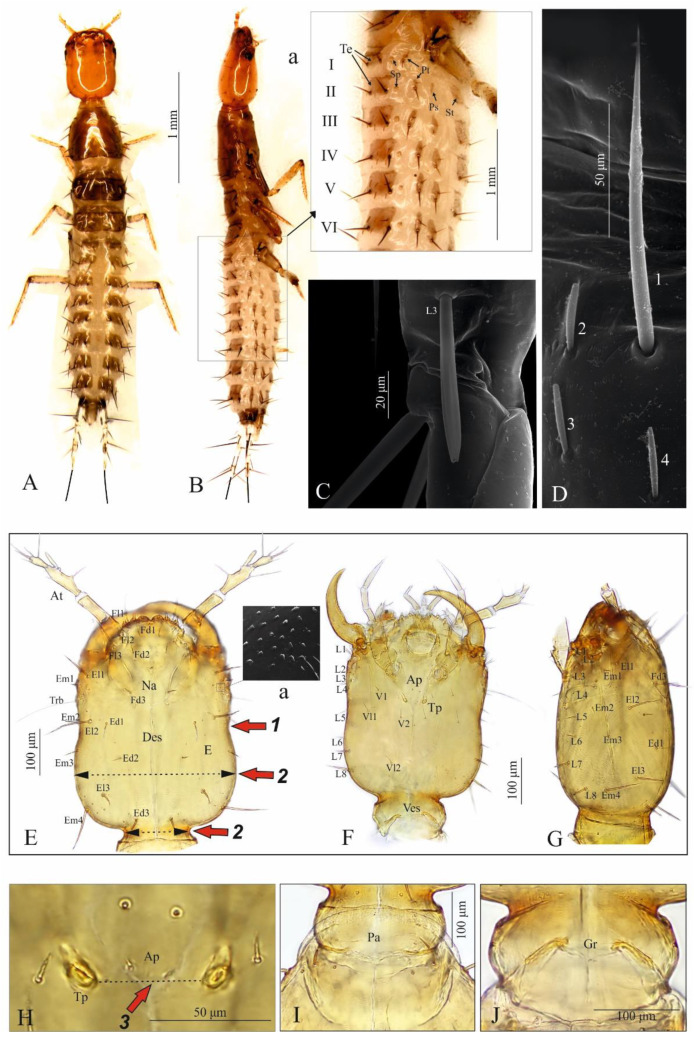
*Pseudomedon obsoletus*, matura larva. (**A**,**B**), entire dorsal (**A**) and lateral (**B**) aspect and magnification of front abdomen (**B**,**a**); (**C**), club-shaped seta (coded: L3) on urogomphus; (**D**), setae on abdominal sternite (coded: 1, long pointed, 2–4, short obtuse); (**E**–**G**), head in dorsal (**E**), ventral (**F**) and lateral (**G**) aspect and microstructure near stemmata (**E**,**a**); (**H**), apotome and tentorial pit region; (**I**,**J**), posterior area of head in dorsal (**I**) and ventral (**J**) aspect. Abbreviations: Ap—apotome, At—antenna, Des—dorsal ecdysial suture, E—epicranial part, Ed—epicranial dorsal seta, El—epicranial lateral seta, Em—epicranial marginal seta, Fd—frontal dorsal seta, Fl—frontal lateral seta, Gr—gular region, L—lateral seta, Na—nasale, Pa—posterior area, Trb—trichobothrium, Tp—tentorial pit, V—ventral seta, Vl—ventral lateral seta, Ves—ventral ecdysial suture, arrows 1–3—characters of genus *Pseudomedon*.

**Figure 3 insects-13-00982-f003:**
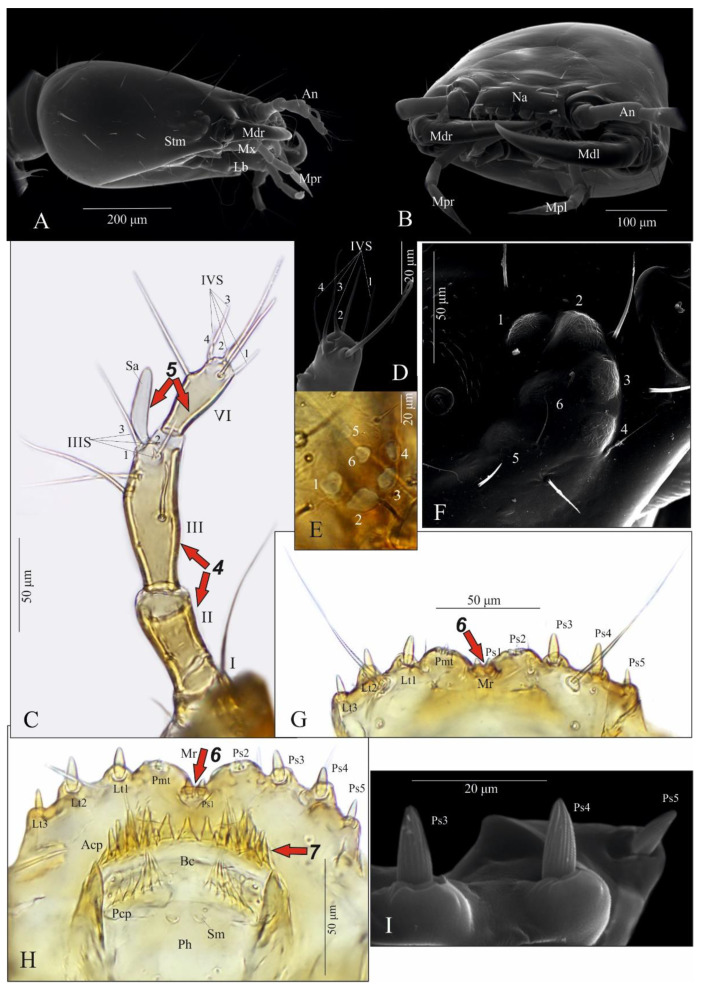
*Pseudomedon obsoletus*, matura larva. (**A**,**B**), head in lateral (**A**) and frontal (**B**) aspect; (**C**,**D**), right antenna, total (**C**) and apical (**D**) view; (**E**,**F**), stemmata left (**E**) and right (**F**) coded 1–6; (**G**–**I**), anterior margin of nasale; (**G**), dorsal view; (**H**), ventral view (epipharynx); (**I**), marginal setae. Abbreviations: I–IV—antennal segments, IIIS1–3—solenidia on antennal segment III, IVS1–4—solenidia on antennal segment IV, Acp—row of cuticular processes along anterior margin of buccal cavity, At—antenna, Bc—buccal cavity, Lb—labium, Lt1-3—lateral teeth, Md—mandible, Mdl—left mandible, Mdr—right mandible, Mpl—left maxillary palp, Mpr—right maxillary palp, Mr—medial region, Mx—maxilla, Na—nasale, Pcp—row of cuticular processes along posterior margin of buccal cavity, Ph—pharynx, Pmt—paramedian tooth, Ps1-5—peg setae, Sa—sensory appendage, Sm—sensillum, St—stemmata, arrows 4–7—characters of genus *Pseudomedon*.

**Figure 4 insects-13-00982-f004:**
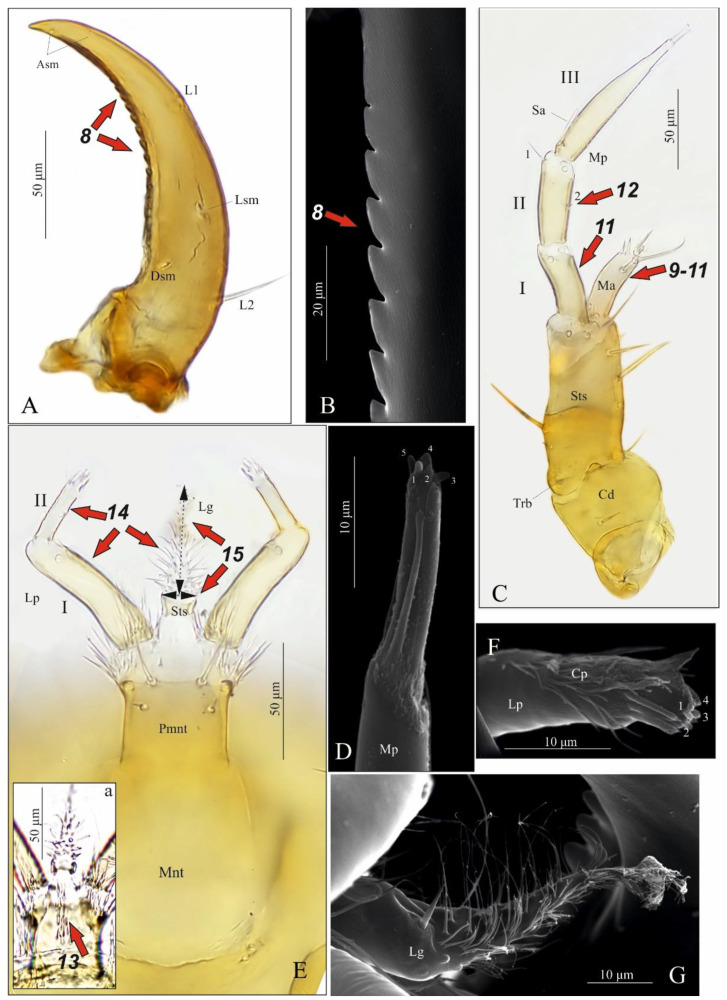
*Pseudomedon obsoletus*, mature larva. (**A**,**B**), right mandible in dorsal aspect, total view (**A**) and magnification of inner margin (**B**); (**C**,**D**), right maxilla in ventral aspect with apical part of maxillary palp (**D**); (**E**,**Ea**,**F**,**G**), labium, ventral aspect (**E**), dorsal aspect (hypopharynx) (**Ea**), apical part of labial palp (**F**) and ligula (**G**) in lateral aspect. Abbreviations: I–II—segments of Lp, I–III—segments of Mp, 1–2—code of setae, 1–4—sensilla on top of Mp, 1–5—sensilla on top of Lp, Cd—cardo, Cp—cuticular processes, Dsm—dorsal sensillum, L1-2—code of mandibular setae, Lg—ligula, Lp—labial palp, Lsm—lateral sensillum, Ma—mala, Mp—maxillary palp, Sa—sensory appendage, Sm—sensillum, Ss—sclerotized strip, Sts—stipes, Trb—trichobothrium, Mnt—mentum, Pmnt—prementum, arrows 8–15—characters of genus *Pseudomedon*.

**Figure 5 insects-13-00982-f005:**
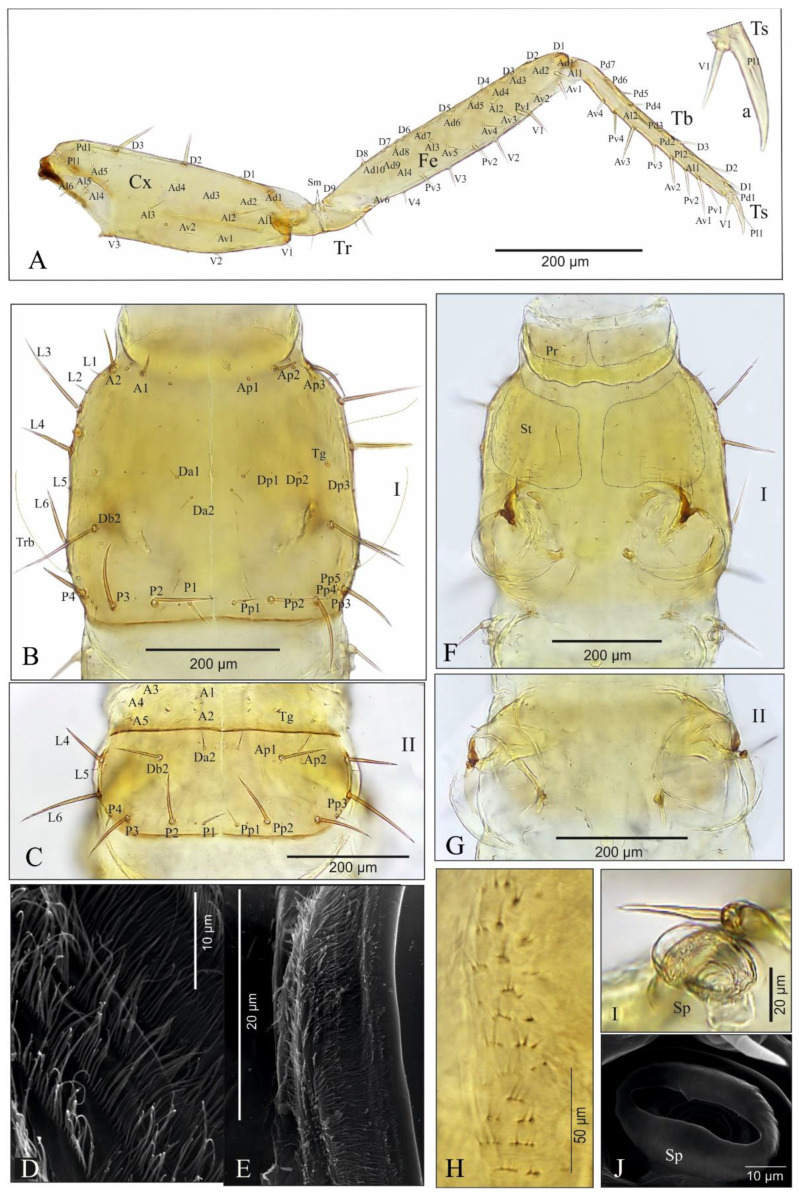
*Pseudomedon obsoletus*, mature larva. (**A**–**J**), thorax; (**A**), right foreleg in frontal aspect; (**B**), pronotum; (**C**), mesonotum; (**D**,**E**), microtrichia of anterior (**D**) region of pronotum and posterior (**E**) region of mesonotum; (**F**), ventral side of prothorax; (**G**), ventral side of mesothorax; (**H**), microstructure of lateral region of thoracic sternite (**I**); (**I**,**J**), thoracic spiracle. Abbreviations: I, II—segments of thorax, A—anterior seta, Ad—anterodorsal seta, Al—anterolateral seta, Ap—anterior pore, Av—anteroventral seta, Cx—coxa, D—dorsal seta, Db—discal seta, row b, Dp—dorsal pore, Fe—femur, L—lateral seta, P—posterior seta, Pd—posterodorsal seta, Pl—posterolateral seta, Pp—posterior pore, Pr—presternum, Pv—posteroventral seta, Sp—spiracle, St—sternum, Tb—tibia, Tg—tergal gland, Tr—trochanter, Trb—trichobothrium, Ts—tarsus, V—ventral seta.

**Figure 6 insects-13-00982-f006:**
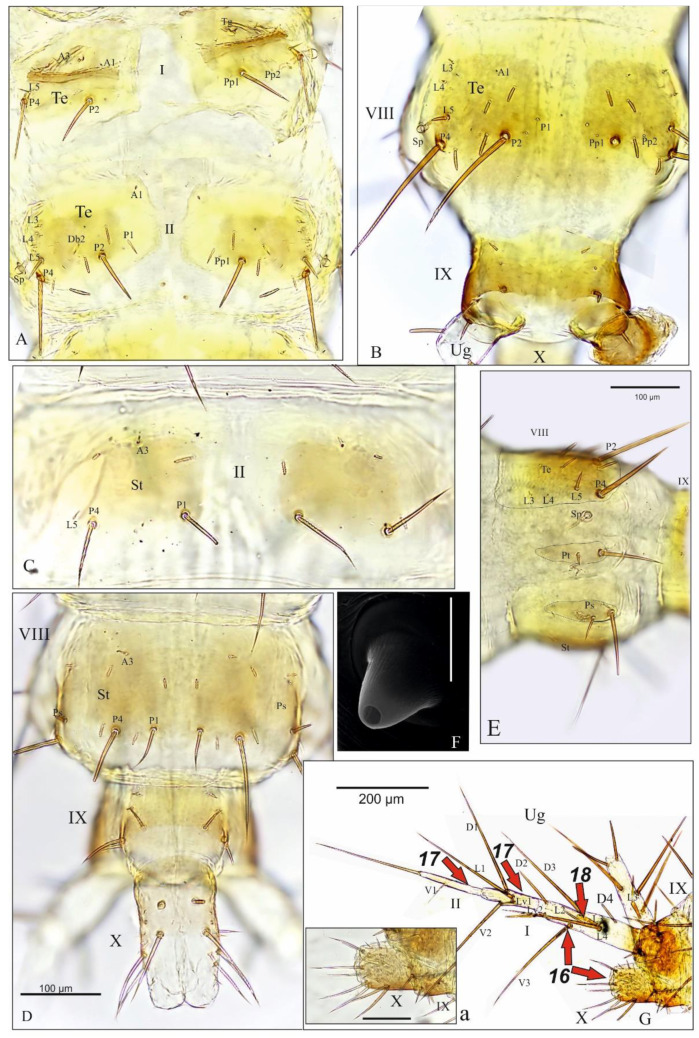
*Pseudomedon obsoletus*, mature larva, abdomen; (**A**), tergites I and II; (**B**), segments VIII–X in dorsal aspect; (**C**), sternite II; (**D**), segments VIII-X in ventral aspect; (**E**), segment VIII in lateral aspect; (**F**), spiracle of segment IV; (**G**,**Ga**), segment IX with urogomphus and segment X (**Ga**) in lateral aspect. Abbreviations: A—anterior setae, D—dorsal seta, Db—discal seta, row b, L—lateral seta, Lv—lateroventral seta, P—posterior seta, Pp—posterior pores, Ps—parasternite, Pt—paratergite, Sp—spiracle, St—sternite, Te—tergite, Ug—urogomphus, V—ventral setae, arrows 16-18—characters of genus *Pseudomedon*.

**Table 1 insects-13-00982-t001:** Known larval morphology of Paederinae species. State of knowledge of morphology, descriptions: very good—descriptions extremely detailed; good—descriptions sufficiently detailed; fair—descriptions moderately informative; poor—descriptions insufficiently informative. State of knowledge of morphology, figures: very good—very well illustrated (the whole structure), usually with SEM photos; good—illustrations sufficient, but without SEM photos; fair—illustrations insufficient; poor—illustrations absent or sketches at best. State of knowledge of morphology, generally: very good—descriptions extremely detailed and well-illustrated, reliable for diagnostics and sufficient for phylogenetic analysis; good—descriptions sufficiently detailed with adequate (sometimes a little insufficient) illustrations, reliable for diagnostics but not fully so for phylogenetic analysis; fair—descriptions moderately informative, possibly with or without adequate illustrations, can be used for diagnostics but not for phylogenetic analysis; poor—descriptions sketchy, mostly without any illustrations or no description with inadequate illustrations, possibly ambiguous even for diagnostic purposes. Abbreviations: abdomen A, antenna An, epipharynx Ep, head H, hypopharynx Hp, labium Lb, labrum Lr, ligula Lg, labial palp Lp, leg L, mandible Md, maxilla Mx, maxillary palp Mp, nasale Na, pygopod Pg, thorax T, urogomphus Ug, all structures AS, diagnostic key characters only DKChO, general habitus GH, state of knowledge SK, selected structures SS, structures ST, total state of knowledge TSK. Symbols: * the table takes account of papers published since Bøving and Craighead [27], earlier ones, being solely of historical interest, were not analysed; ** after Paulian [19]?—difficult to assess because only a few key diagnostic characters were given.

Species	Descriptions			Illustrations			TSK	References *
	GH	ST	SK	GH	ST	SK		
Tribe: Pinophilini								
Pinophilini Nordmann, 1837	absent	general	-	absent	H, An, Md, Mx, Lb, L, T, Ug	poor	poor	[19], pp. 184–186
Tribe: Lathrobiini *sensu novo*								
*Astenus* sp. Dejean, 1833	absent	DKChO	fair	absent	absent	-	fair	[12], p. 54
	absent	DKChO	fair	absent	absent	-	fair	[13], p. 310
*A. ?angustatus* (Paykull, 1789)	absent	absent	-	absent	Na, Mx	poor	poor	[12], p. 53
*A. altivagans* Bernhauer, 1939	present	SS	poor	absent	H, An, Md, Mx, T, L, A4, Ug	poor	poor	[16], pp. 358–359
*Deroderus* sp. Sharp, 1886	absent	AS	good	absent	Ep	poor	good	[3], pp. 2547, 2550–2553
*Domene* (*Canariomene*) *alticola* Oromi & Hernández, 1986	present	AS	fair	present	An, Na, Md, Mx, Lb, T, L, Ug	fair	fair	[28], pp. 89–93
*D.* (*C.*) *benahoarensis* Oromi & Martin, 1990	present	AS	fair	present	H, An, Na, Md, Mx, Lb, T, L, A3, Ug	fair	fair	[28], pp. 80–84, 93
*D.* (*C*) *vulcanica* Oromi & Hernández, 1986	present	AS	fair	present	H, An, Na, Md, Mx, Lb, T, L, Ug	fair	fair	[28], pp. 85–88, 93
*Eustilicus**fasta* (Sanderson, 1947)	present	AS	very good	present	H, Lr, Md, Mx, Lb, T, L, A, Ug	good	very good	[3], pp. 2544–2550
*E. tristis* (Melsheimer, 1844) = *Stilicolina* *tristis* (Melsheimer, 1846)	absent	SS	poor	absent	H, An, Mx, Lb, T	poor	poor	[29], pp. 6–7
*Lathrobium* sp. Gravenhorst, 1802	absent	DKChO	fair	absent	absent	-	fair	[12], pp. 52
	present	DKChO	fair	absent	absent	-	fair	[13], pp. 310
	present	SS	poor	absent	H, Ug	poor	poor	[19], pp. 199–200
	absent	-	-	absent	H, Na, Mx, Lb	poor	poor	[12], pp. 51
*Lathrobium* (*Glyptomerus*) *alzonai* Capra & Binaghi, 1938	present	SS	fair	absent	An, Na, Mx, Lb, L, Ug	poor	fair	[30], pp. 29–33
*L.* (*G.*) *freyi* Koch, 1938	present	SS	fair	present	An, Na, Mx, Lb, L, Ug	poor	fair	[31], pp. 219–221
*Lathrobium* (*G.*) *cavicola* (Müller, 1856) = *Glyptomerus cavicola* Müller, 1856	present	general	poor	absent	H, An, Mx, Lb, L	poor	poor	[19], pp. 202–203
	absent	DKChO	fair	absent	An, Lr, Mx, Lb, Ug	poor	fair	[30], pp. 31–33
	absent	DKChO	fair	absent	H, An, Mx, Lb, L, **	poor	fair	[11], pp. 36–37, 87
*L.* (*Lathrobium*) *brunnipes* (Fabricius, 1792)	absent	-	-	absent	H, Na, Mx, Lb	poor	poor	[12], pp. 51
	absent	DKChO	fair	absent	Md, Lr, A9,10	poor	fair	[11], pp. 35, 39
	absent	absent	-	absent	Na, Mx	poor	poor	[13], pp. 321
*L.* (*L.) elongatum* (Linnaeus, 1767)	absent	absent	-	absent	An, Ug	poor	poor	[13], pp. 321, 322
	absent	DKChO	fair	absent	Lr, Md, A9,10	poor	fair	[11], pp. 35, 39
	present	SS	poor	absent	H, Mx, L, Ug	poor	poor	[19], pp. 200–201
*L.* (*L.*) *fulvipenne* (Gravenhorst, 1806)	absent	DKChO	fair	absent	Lr, Md	poor	fair	[11], pp. 35, 39
*L.* (*L.*) *geminum* Kraatz, 1857 = *L. rufescens* Motschulsky, 1860	absent	DKChO	fair	absent	An, Lr, Md, Mx, Lb, Ug	poor	fair	[11], pp. 35, 39
*L.* (*L.*) *grande* (LeConte, 1863)	present	AS	good	absent	H, An, Na, Md, Mx, Lb, T, L, A5,9,10, Ug	fair	good	[9], pp. 146–149
*L.* (*L.*) *lineatocolle* Scriba, 1859	present	AS	very good	present	AS	very good	very good	[7], pp. 2–14
*Lobrathium badium* (Cameron, 1924) = *Lathrobium* *badium* Cameron, 1924	present	SS	poor	absent	H, Lb, T, L, A3	poor	poor	[19], pp. 201–202
*L. emarginatum* (Watrous, 1981) = *Lathrobium* *emarginatum* Watrous, 1981	absent	?	-	absent	absent	-	-	[9], p. 150
*Lithocharis* sp. Dejean, 1833	absent	DKChO	fair	absent	absent	-	fair	[13], p. 310
	present	SS	poor	absent	absent	-	poor	[19], pp. 197, 199
*L. nigriceps* Kraatz, 1859	absent	absent	-	absent	H	poor	poor	[19], pp. 198–199
*L. vilis* Kraatz, 1859	absent	DKChO	fair	absent	H, An, Lr, Lb, A10 **	poor	fair	[11], p. 38
	absent	absent	-	absent	H, An, Lr, Mx, Lb, L, T, A3	poor	poor	[19], p. 198
*Medon* sp. Stephens, 1833	absent	DKChO	fair	absent	absent	-	fair	[12], p. 54
*M. johni* Blackwelder, 1943	absent	AS	good	absent	Mx, T	good	good	[3], pp. 2548–2549, 2553–2554
*Myrmecosaurus solenopsidis* Wasmann, 1909	present	SS	poor	absent	H, Mx, Lb, L, Ug	poor	poor	[19], pp. 191, 194,
*Rugilus* sp. Leach, 1819 = *Stilicus* Berthold, 1827	absent	DKChO	fair	absent	absent	-	fair	[12], p. 54
		DKChO	fair	absent	absent	-	fair	[13], p. 310
	absent	AS	good	absent	Lr, Ep, Lb	good	good	[3], pp. 2547–2548, 2554
*R*. (*Rugilus*) *orbiculatus* (Paykull, 1789) = *Stilicus orbiculatus* Paykull, 1789	absent	absent	-	absent	H, Mx, Lb	poor	poor	[12], p. 53
	absent	DKChO	-	absent	Na, Ug	poor	poor	[13], p. 322
	absent	DKChO	fair	absent	H, Mx, Ug **	poor	fair	[11], pp. 33, 38
	present	SS	poor	absent	H, An, Mx, Lb, T, L, A3, Ug	poor	poor	[19], pp. 195–197
*R. (R.) rufipes* Germar, 1836	absent	DKChO	fair	absent	Lr, Ep, Md, Mx, Lb, L, Ug	poor	fair	[11], pp. 34, 38
	present	AS	good	present	H, Lr, An, Na, Md, Ep, Mx, Lb, T, A, Ug	good	good	[10], pp. 234–242
*Stilocharis* sp. Sharp, 1886	absent	SS	good	absent	H, L, Ug	poor	good	[3], pp. 2546, 2552–2553
*Tetartopaeus quadratus* (Paykull, 1789)	present	AS	very good	present	AS	very good	very good	[7], pp. 2–14
Tribe: Paederini *sensu novo*								
*Ochtephilum* sp. Stephens, 1829 = *Cryptobium* sp. Mannerheim, 1830	absent	DKChO	fair	absent	H, Na, An, Mx, Lb	poor	fair	[12], pp. 51–52
	absent	DKChO	fair	absent	absent	-	fair	[13], p. 310
*O. fracticorne* (Paykull, 1800) = *C. fracticorne* (Paykull, 1798)	absent	DKChO	fair	absent	H, An, Md, Lb, Ug **	poor	fair	[11], pp. 37–38
	present	SS	poor	absent	H, An, Md, Mx, Lb, T, A4, Ug	poor	poor	[19], pp. 192–193
*Paederus* sp. Fabricius, 1775	absent	DKChO	fair	absent	absent	-	fair	[12], p. 52
	absent	DKChO	fair	absent	absent	-	fair	[13], p. 310
*P.* (*Eopaederus*) *basalis* Bernhauer, 1914	absent	SS	poor	absent	Lr	poor	poor	[19], p. 191
	absent	SS	poor	absent	absent	-	poor	[16], p. 355
	absent	DKChO	?	absent	-	-	?	[15], p. 188
*P*. (*E.*) *caligatus* Erichson, 1840	absent	SS	poor	absent	H, Mx	poor	poor	[19], pp. 187–188
	absent	DKChO	poor	absent	H **	poor	poor	[11], pp. 33–38
	absent	DKChO	?	absent	absent	-	?	[15], p. 188
*P. (Heteropaederus) fuscipes* Curtis, 1826	absent	SS	poor	absent	An, Mx, Lb, L, Ug	poor	poor	[19], pp. 189, 190
	present	SS	poor	absent	absent	-	poor	[14], pp. 200–201
	present	DKChO	fair	absent	A6	poor	fair	[32], pp. 29–31
	?	?	?	?	?	?	?	[18], ?
	present	?	?	present	H, An	poor	?	[17], ?
	absent	DKChO	poor	absent	absent	-	poor	[11], pp. 38
*P.* (*H*.) *alfierii* Koch, 1934	present	AS	good	present	H, An, Na, Md, Ep, Mx, Lb, L, T, A5, Ug	fair	good	[8], pp. 129–136
*P. (Poedermorphus) littoralis* Gravenhorsth, 1802	absent	absent	-	absent	Na, Mx, Lb	poor	poor	[12], p. 51
	present	SS	fair	absent	H, Lr, An, Md, Mx, Lb, Hp, T	fair	fair	[15], pp. 184–189
	present	AS	very good	present	AS	very good	very good	[7], pp. 2–10, 12–14
*P.* (*Paederus*) *riparius* (Linneus, 1758)	present	absent	-	absent	Na	poor	poor	[13], p. 322
	absent	absent	-	present	H, Hp, A	poor	poor	[27], p.p 117, 123
*P.* (*P.*) *sabaeus* Erichson, 1840	present	SS	poor	absent	H, An, Md, Mx, Lp, L, A3, Ug	poor	poor	[19], pp. 190, 191
	absent	SS	poor	absent	absent	-	poor	[16], p. 355
	absent	DKChO	?	absent	absent	-	?	[15], p. 188
*P. columbinus* Laporte, 1835	present	SS	poor	present	H, Lr, An, Md, Mx, T, A7-9	fair	poor	[33], pp. 43–47
*P. conspicuous* Erichson, 1840	absent	DKChO	poor	absent	absent	-	poor	[33], p. 42
*P. hintzi* Bernhauer, 1915 = *P. steelei* Bernhauer, 1939	present	SS	poor	absent	H, Mx, L, A4, Ug	poor	poor	[16], pp. 354–355
*P. machadoi* Sheerpeltz, 1965	absent	DKChO	poor	absent	absent	-	poor	[33], p. 42
*P. marcuzzii* Sheerpeltz, 1965	absent	DKChO	poor	absent	absent	-	poor	[33], p. 42
*P. signaticornis* Sharp, 1886	absent	DKChO	poor	absent	absent	-	poor	[33], p. 42
*P. tempestivus* Erichson, 1840	absent	DKChO	poor	absent	absent	-	poor	[32], p. 31
	absent	DKChO	-		absent	-	-	[15], p. 188
*Paederidus algiricus antoinei* (Koch, 1937) = *Paederus* *antoinei* Koch, 1937	absent	SS	poor	absent	An, Mx, Lb, L, Ug	poor	poor	[19], pp. 188–189
	absent	DKChO	?	absent	absent	-	?	[15], p. 188
*Paederidus rubrothoracicus* Goeze, 1777	absent	SS	poor	absent	absent	-	poor	[19], p. 188
	absent	DKChO	?	absent	absent	-	?	[15], p. 188
*P. rubrothoracicus carpathicola* (Scheerpeltz, 1957)	present	AS	very good	present	AS	very good	very good	[6], pp. 41, 44–50, 52–55
*P. ruficollis* (Fabricius, 1777)	present	AS	fair	absent	H, Lr, A6	poor	fair	[32], pp. 27–31
	present	AS	very good	present	AS	very good	very good	[6], pp. 41–50, 53–55
	absent	DKChO	?	absent	absent	-	?	[15], pp. 186, 188

**Table 2 insects-13-00982-t002:** Some measurements of the studied specimens of larval instars of *Pseudomedon obsoletus*. Abbreviations: L1-2—larval instars; N—number of specimens; RM—range of measurement; M—mean; SD—standard deviation.

Measurements (mm)	Instar/N					
	L1/8			L2/7		
	RM	M	SD	RM	M	SD
Body length	1.32–3.10	2.50	0.69	3.30–3.99	3.56	0.40
Thorax length	0.54–1.00	0.88	0.18	1.10–1.50	1.32	0.19
Pronotum width	0.26–0.36	0.33	0.07	0.36–0.44	0.39	0.02
Head width	0.30–0.36	0.33	0.03	0.41–0.43	0.42	0.01
Head length	0.30–0.53	0.43	0.09	0.53–0.57	0.55	0.02

**Table 3 insects-13-00982-t003:** Larval characters (in order of power) common to Lathrobiini *sensu novo* and Paederini *sensu novo*. Abbreviations: Acp—row of cuticular processes along anterior margin of buccal cavity, Ar—article, At—antennal, Lg—ligula, Lp—labial palp, Ma—mala, Mp—maxillary palp, Na—nasale, Pmt—paramedian tooth, Rw—ring of wrinkled cuticula, Sa—sensory appendage, Sts—stipes; * at least 2× as narrow as Ar I of Mp, ** at most only slightly narrower than Ar I of Mp.

Character	i. Oral Opening	ii. Seta on At ArII	iii. Length of At Sa	iv. Number of Teeth on Na; v. Gap between Pmt	vi. row of Acp of Epipharynx; Tuft of Acp Medially	vii. Rw; viii. sensilla on ArII of Mp and Lp	ix. Shape of Sts	x. Shape of Lg	xi. Shape of Ma
Paederini sensu novo	wide	present	short	4; present	incomplete; present	present; absent	trapeziform	conical	slim *
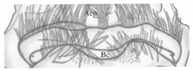	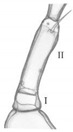	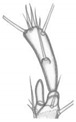	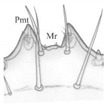	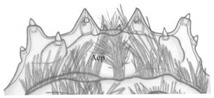	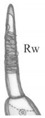	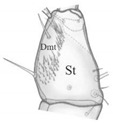	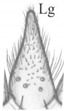	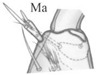
Lathrobiini sensu novo	narrow	absent	long	7–9; absent	complete; absent	absent; present	rectangular	bulbif-form	robust **
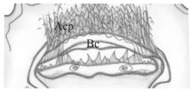	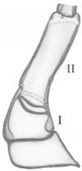	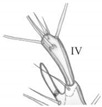	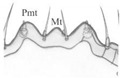	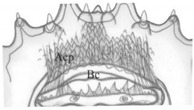	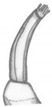	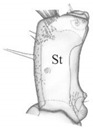	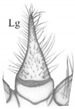	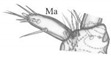

**Table 4 insects-13-00982-t004:** Comparative characters of the larva of *Pseudomedon* and some previously described larvae of Paederinae. Abbreviations: Acp—row of cuticular processes along anterior margin of buccal cavity; Ap—apotome; Ar—article; Asg—antennal segment; Bm—bunch of microtrichia at the mandible base; Hr—height ratio; Lg—ligula; Lr—length ratio; Lwr—length-to-width ratio; Mt—median tooth; Ncp—number of cuticular process; NmaS—number of macro setae; NmiS—number of micro setae; Npgs—number of peg setae; Ns—number of setae; Nt.Am.—number of well-developed teeth, anterior margin; Mss—match-shaped setae; Pg—pygopod (abdominal segment X); Pmnt—prementum; Pmt—paramedian tooth; Rw—ring of wrinkled cuticula; Sa—sensory appendage; Sap—apical seta; Sts—sclerotised transversal strip; Tp—tentorial pits; Tsn—terminal sensilla.

Character	Tribe/Subtribe/Genus/Species					
	LATHROBIINI				PAEDERINII	
	Medonina	Lathrobiina		Stilicina	Paederina	
	*Pseudomedon*	*Lathrobium*	*Tetartopeus*	*Rugilus* sp.,	*Paederus*	*Paederidus*
	*obsoletus*	*lineatocolle*	*quadratus*	*R. rufipes*	*littoralis*	*ruficollis*
Body length	3.3–4.0	4.1–5.2	4.5–4.9	5.5–7.5	5.0-5.1	5.2–7.3
Mss	present	present	present	?	absent	absent
HEAD						
Width	0.30–0.36	0.8–0.9	0.71–0.75	?	0.82–0.8	0.74–0.83
Length	0.30–0.53	0.9–1.2	0.9–1.0	?	0.73–0.94	?
Shape in outline	rectangular	U-shaped	semi-U-shaped	rectangular	trapezoidal	semicircular
Wr: head to neck	2.5:1	3.6:1	1.8:1	3.1:1	1.9:1	1.9:1
NmaS: epicranial part	10	14	16	20	24	22
NmiS: dorsal and ventral side	absent	numerous	numerous	numerous	absent	absent
Ap	present	present	present	absent	present	absent
Ap: position with respect to Tp	extending to Tp	reaching Tp	reaching Tp	-	extending to Tp	-
ANTENNA						
Asg Ar II: seta	absent	absent	absent	absent	present	present
Ar III: shape of Sa	slightly curved	slightly curved	slightly curved	slightly curved	erect	erect
Lr: Sa and Ar IV	1:1.2	1:1.5	1:1	1:1.5	1:2.9	1:2.1
Ar II-IV: Lwr	7.4×, 7.3×, 5.2×	4.1×, 3.6×, 3.3×	3.1×, 3.5×, 2.6×	3.7×, 4×, 4.3×	3.3×, 3.9×, 3.8×	2.9×, 3.6×, 3.3×
Lwr: Ar I-IV	1:2.7:4.1:2.4	1:4.7:4.1:2.3	1:2.0:2.3:1.1	1:4.4:4.4:2.7	1:4.1:4.4:3.0	1:2.7:3.3:2.0
NASALE						
Nt.Am	8	7	7	8	4	4
Mt: anterior margin	absent	present	present	absent	absent	absent
Gap between Pmt	absent	absent	absent	absent	present	present
Npgs	10	10	10	8	10	8
EPIPHARYNX						
Row of Acp	complete	complete	complete	complete	incomplete	incomplete
Acp: Ncp	22–25	about 25	about 50	?	about 120	about 100
Microstructure	absent	present	present	present	absent	absent
MANDIBLE						
Inner margin	serrate	smooth	serrate	serrate	serrate	serrate
Bm	absent	absent	absent	?	present	present
MAXILLA						
Stipes: shape	rectangular	rectangular	rectangular	rectangular	tapering to the top	tapering to the top
Stipes: Lwr	1.9×	2.2×	2.0×	1,8×	1.4×	1.4×
Mala: Lwr	3.1×	3.5×	2.6×	3.8×	2.8×	3×
MAXILLARY PALP						
Lr: mala and Ar I	1:1	1:1.3	1:1.6	1.1:1	1:2.0	1:2.2
Wr: mala and Ar I	1:1.2	1:1	1:1.5	1:1.4	1:3	1:3.2
Ar I-III: Lwr	5.8×, 7×, 13.4×	4.2×, 4.1×, 8.6×	4.1×, 4.2×, 11×	2.8×, 5.1×, 9.6×	1.9×, 2.8×, 5.8×	1.6×, 2×, 4.1×
Lr of Ar I-III	1:1.1:2.4	1:1:1.6	1:1:2	1:1.5:2	1:1.5:2.4	1:1.6:2.4
Ar II: seta 2: length/position	short/ at half-length	short/ at half-length	short/ at half-length	short/ close to top	long/basally	long/basally
Tsm	long	long	long	long	rudimentary	rudimentary
Ar III: Rw	absent	absent	absent	absent	present	present
HYPOPHARYNX						
Microtrichia: number/position	over a dozen/central	numerous/ anterocentral	numerous/ anterocentral and lateral	?	numerous/ anterocentral	numerous/ anterocentral
LABIUM						
Lg: Lwr	3.1×	2.4×	2.3×	?	2.1×	1.8×
Lg: shape	bulbiform	bulbiform	bulbiform	bulbiform	conical	conical
Sts	present	present	present	present	absent	absent
Pmnt: shape	extended anteriorly	extended anteriorly	rectangular	extended anteriorly	trapeziform	trapeziform
LABIAL PALP						
Lg and Ar I: Lr	1:1.3	1:1.4	1:1.6	1:1.3	1:1	1:1
Ar I-II: Lwr	3.4×, 2.6×	3.9×, 4×	2.9×, 3×	5×, 6.8×	4×, 7×	3×, 5×
Ar I-II: Lr	2.1:1,	1.7:1	1.9:1	1.6:1	1.5:1	1.3:1
Ar II: Rw	absent	absent	absent	absent	present	present
Ar II: shape	curved	curved	curved	erect	erect	erect
Tsm	long	long	long	long	rudimentary	rudimentary
UROGOMPHUS						
Ar I: Ns	12	14	14	11	?	10
Ar I, Ar II, Sap: Lr	2.5:1:1.6	3.9:1:2.5	4.9:1:1.7	1.5:1:1.5	?	2.4:1:1.4
Lwr Ar I	6.5×	8.5×	11.8×	5.5×	?	7×
Ug and Pg: Lr	4.4:1	3.3:1	6.5:1	?	?	6.7–7.1:1
References	present study	[7]		[3,10]	[6]	

## Data Availability

Not applicable.

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
