# Peer review of "Synthesis of Current Knowledge of the Morphology of the Larval Stages of Paederinae (Coleoptera; Staphylinidae), with a First Insight into the Mature Larva of Pseudomedon Mulsant & Rey, 1878, in the Light of a New Systematic Division"

_insects, 2022, doi:10.3390/insects13110982_

Round 1

Reviewer 1 Report

This is a nice paper on the difficult subject. It has considerable new information, good review of the previously published data and everything is presented in great detail. Perhaps the very detailed presentation is also an enemy of this paper, because the details (some of them are important, some of them are not so) blur clear main messages. For example, it would be good to give clear simple larval diagnoses of the tribes Lathrobiini and Paederini after the diagnosis of Paedetinae. It would be good to highlight potential synapomorphies of both tribes. The English language and style of some sentences needs some improvement. For example, sentences like the one on the lines 342-345 on the page 10 is confusing. (Also, it is better to replace the bold words like "useless" etc. by something milder and more exact like, for example, - "poorly informative" for... etc.).  Small corrections - Zyla on page 10, line 322 should be Zyla et al., the title for the key to genera should be "Key to genera of the known larvae...". The first section of the "Results" on page 7 contain information from the journal guidelines. It must be left there by mistake. I admit that I only quickly read the paper to complete the review, because the journal editorial office sends reminders too often and wants the review a bit too soon, which is hard to do under other commitments. This paper would benefit from a thorough reviewer. Overall I recommend publication as it is a valuable contribution.

Author Response

"Perhaps the very detailed presentation is also an enemy of this paper, because the details (some of them are important, some of them are not so) blur clear main messages".

Response: We agree that too detailed presentation may blur the main message. However, at this preliminary stage of research on Paederiane larvae, it seems expedient to present a set of characters (in detail). This will facilitate, in the near future, to conduct a comparative analysis of the larval morphological features including genera not yet studied (or poorly studied): Lithocharis, Ochthephilum, Sunius, etc. and identify these crucial diagnostic features at different taxonomic levels.

"For example, it would be good to give clear simple larval diagnoses of the tribes Lathrobiini and Paederini after the diagnosis of Paedetinae". 

Response:  That larval diagnoses of both tribes (Lathrobiini and Paederini) are presented in Table 3 (line 373) together with the figures. Due to the fact that the diagnosis requires a broader commentary based on the literature (mainly from Frania, 1985), it seems that including it in the chapter "discussion" is a far better solution than transferring it after diagnosis of Paederinae.

"It would be good to highlight potential synapomorphies of both tribes".

Response: Providing potential synapomorphies at this early stage of research on the larvae of Paederinae, when only five genera are described, does not seem necessary. The more so as a thorough morphological and phylogenetic study based on the larval characters is planned in the near future.

"The English language and style of some sentences needs some improvement. For example, sentences like the one on the lines 342-345 on the page 10 is confusing. (Also, it is better to replace the bold words like "useless" etc. by something milder and more exact like, for example, - "poorly informative" for... etc.)". 

Response: accepted and corrected; the English language and style of the entire text has been proofread by native speaker Peter Senn.

"Small corrections - Zyla on page 10, line 322 should be Zyla et al., the title for the key to genera should be "Key to genera of the known larvae...". The first section of the "Results" on page 7 contain information from the journal guidelines. It must be left there by mistake". 

Response: all accepted and corrected

Reviewer 2 Report

The manuscript is well written and provides a great deal of morphological information on the Paederinae.  The new morphological information provided in this text is extremely valuable to scientists, and more research on taxonomic identification should be conducted.  

Specific comments:

Line 42-45 could be reworded.  ...characters of adults, as lesser extent on larval stages, and very recently....

Line 72:  Change "absolutely useless" to "has no value".

No other editorial suggestions.

Author Response

Specific comments:

Line 42-45 could be reworded.  ...characters of adults, as lesser extent on larval stages, and very recently....

Line 72:  Change "absolutely useless" to "has no value".

Response: accepted and corrected.